# STSBench: A Large-Scale Dataset for Modeling Neuronal Activity in the Dorsal Stream of Primate Visual Cortex

**Ethan B. Trepka**[1], **Ruobing Xia**[2], **Shude Zhu**[2,3], **Sharif Saleki**[2],
**Danielle Abreu Lopes**[2,3], **Stephen J. Niño Cital**[2,3], **Konstantin F. Willeke**[4],
**Mindy Kim**[2,3], **Tirin Moore**[2,3,*]

[1]Neuroscience Interdepartmental Program, Stanford University, Stanford, CA
[2]Department of Neurobiology, Stanford University, Stanford, CA
[3]Howard Hughes Medical Institute, Stanford University, Stanford, CA
[4]Department of Opthalmology, Stanford University, Stanford, CA
[*]Corresponding author, tirin@stanford.edu

## Abstract

The primate visual system is typically divided into two streams — the ventral stream, responsible for object recognition, and the dorsal stream, responsible for encoding spatial relations and motion. Recent studies have shown that convolutional neural networks (CNNs) pretrained on object recognition tasks are remarkably effective at predicting neuronal responses in the ventral stream, shedding light on the neural mechanisms underlying object recognition. However, similar models of the dorsal stream remain underdeveloped due to the lack of large scale datasets encompassing dorsal stream areas. To address this gap, we present STSBench, a dataset of large-scale, single neuron recordings from over 2,000 neurons in the superior temporal sulcus (STS), a nearly 50-fold increase over existing dorsal stream datasets, collected while Rhesus macaques viewed thousands of unique, natural videos. We show that our dataset can be used for benchmarking encoding models of dorsal stream neuronal responses and reconstructing visual input from neural activity.

## 1 Introduction

A principal goal of systems neuroscience is to characterize the map between external stimuli and neuronal responses [1]. This question has often been studied by probing neurons with simple parametric stimuli to explain the relationship between stimulus parameters and neuronal responses [2, 3]. In particular, these studies have proven instrumental in shaping our understanding of single-neuron response properties throughout the primate visual system [e.g. 4] as well as its functional organization [e.g. 5]. However, these classic approaches fail to adequately capture complex neuronal responses to natural scenes [6], particularly in higher-level visual areas where neurons encode increasingly abstract and nonlinear features [7].

Visual processing within the primate visual system is accomplished by functionally specialized, quasi-separable neural circuits. The output of the primate retina consists of distinct classes of ganglion cells distinguished by their relative specialization for spatiotemporal or object vision [8]. The vast majority of retinal output is transmitted to primary visual cortex (V1) via anatomically distinct layers of the dorsal lateral geniculate nucleus in which the above specializations remain largely segregated. That segregation continues in V1 and to a large extent in V2. Beyond these areas lie many additional visual representations extending dorsally into the parietal lobe where neurons specialize in spatiotemporal

39th Conference on Neural Information Processing Systems (NeurIPS 2025) Track on Datasets and Benchmarks.

vision and ventrally into the temporal lobe where neurons specialize in object vision (**Figure 1a**) [9]. For example, whereas neurons in ventral visual areas tend to be more selective to shape and color (e.g. area V4) [10–12] and to foveal stimuli (e.g. inferotemporal cortex) [13], neurons in dorsal areas are more selective to motion and spatial processing (e.g. area MT) [14, 15]. More crucially, selective impairments of object identification follow damage to ventral visual areas whereas more spatial deficits follow damage to dorsal visual areas [16].

In recent years, convolutional neural networks (CNNs) have been applied to predict neuronal response to natural images in ventral stream areas such as V4 and the inferior temporal (IT) cortex with considerable success [7, 17–20]. These advances were accelerated by the release of a suite of large-scale datasets and benchmarks for predicting ventral stream neuronal responses to natural images, including BrainScore [21, 22], MacaqueITBench [23], and the Things Ventral Stream Dataset (TVSD) [24], among others [e.g. 25]. In contrast, datasets and models of dorsal stream neuronal responses to naturalistic stimuli remain scarce. To date, the largest such dataset contains 45 neurons recorded from area MT [26, 27] which is orders of magnitude smaller than ventral stream datasets such as TVSD that contain thousands of neurons [24]. The limited scale of existing datasets has constrained the development of deep learning-based models for predicting neuronal responses in the dorsal stream [but see 28].

In the past several years, high-channel-count electrophysiological recording devices such as Neuropixels probes have transformed neuroscience by enabling simultaneous recordings from large, densely localized populations of neurons anywhere in the brain. These recording probes were deployed initially in rodents [29], and subsequently in primates [30]. The capabilities provided by such probes have already led to several novel discoveries [31, 32]. Short-length (10 mm) probes were first used to record neurons in both human and nonhuman primates (NHPs), allowing access to superficial targets. More recently, Neuropixels probes were adapted for greater suitability in primates by extending the probe length in order to achieve large-scale recordings in deep structures including visual areas located deep within the convolutions of the posterior visual cortex, such as the superior temporal sulcus (STS) [33]. Neuropixels probes are thus ideally suited to build large datasets from the entirety of the primate visual system including both the dorsal and ventral streams. Here, we leveraged these probes to address the relative lack of data from the primate dorsal stream by recording from thousands of neurons in area MT/MST in the STS. We recorded neuronal activity while monkeys viewed natural videos, and used this large-scale dataset to develop new encoding and reconstruction models for the dorsal stream (**Figure 1b-e**).

Our main contributions in this paper are: *(i)* We release STSBENCH, a dataset of single neuron recordings in the STS with 2,244 neurons recorded while monkeys viewed ∼4,500 unique natural videos. *(ii)* We use STSBENCH to benchmark an extensive suite of encoding models, and identify gaps in current models of visual processing in MT/MST. *(iii)* We use a neural-conditional latent diffusion model for reconstructing visual stimuli from neural activity, and demonstrate successful reconstructions on STSBENCH and TVSD.

## 2    Related works

**Neural network encoding models of dorsal and ventral visual cortex.**    A landmark study by Yamins et al. [7] found that CNNs trained on object recognition tasks are highly predictive of neuronal activity in inferior temporal (IT) cortex. Subsequent work demonstrated that the hierarchy of layers in CNNs aligns with the hierarchy of ventral visual areas, with early layers predictive of V1 and later layers predictive of V4 and IT [17, 34]. These models have provided insight into the neuronal mechanisms underlying object recognition, including the functional organization of feature-selectivity in V4 [19] and face-selectivity in IT [35].

Although the functional properties of neurons in the dorsal stream have been extensively investigated using parametric stimuli, there have been comparatively few studies that used naturalistic stimuli. In a pioneering study, Nishimoto and Gallant [26] introduced a model for predicting neuronal responses in area MT to natural videos that consists of a bank of 3D Gabor filters convolved with the video followed by a linear readout. Mineault et al. [28] compared this model to 3D ResNets, trained either on action recognition (ResNet3D-18) or self-motion estimation (DorsalNet) tasks, and concluded that the dorsal stream is optimized for 'self-motion estimation'. A separate line of work has applied these approaches to functional magnetic resonance imaging (fMRI) data, which captures activity in both

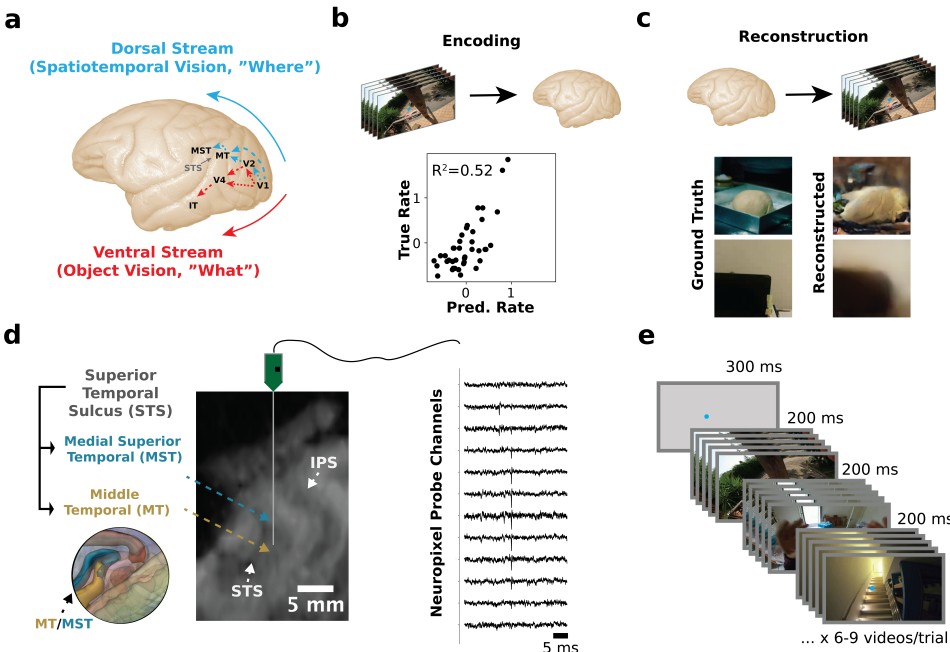

Figure 1: Summary of STSBENCH. **(a)** Diagram of dorsal and ventral visual streams in the macaque brain. The dorsal stream includes the middle temporal area (MT), and the medial superior temporal area (MST), located in the superior temporal sulcus (STS). The ventral stream includes visual area 4 (V4), and inferior temporal cortex (IT). **(b)** Encoding models map from stimuli to neural activity. Plot displays example encoding model results for a neuron in STSBENCH. **(c)** Reconstruction models map from neural activity to stimuli. Images show example reconstruction model results for neurons in the ventral stream (top) and dorsal stream (bottom). **(d)** (Left) Example probe trajectory overlayed on an MR image of the STS from monkey T. Inset displays 3D reconstruction of medial superior temporal (MST; blue) and middle temporal (MT; gold) areas from the anatomical MRI in monkey A. (Right) Extracellular voltage traces recorded from sample channels on the Neuropixels probe. **(e)** Diagram of the video fixation task.

dorsal and ventral visual stream areas [36, 37]. While this line of work has provided valuable insight into large-scale cortical representations, the limited spatial and temporal resolution of fMRI makes it unsuitable for modeling single-neuron activity, which is the focus of STSBENCH.

**Neural network reconstruction models for generating stimuli from neural activity.** The task of reconstructing visual stimuli from neural activity has been extensively studied in the fMRI literature, spurred by the public release of large scale fMRI natural image datasets. A wide range of models have been proposed for this task, including Bayesian decoders [38] generative adversarial networks (GANs) [39], and conditional latent diffusion models [40]. STSBENCH, complements this body of work by providing neural data from a fundamentally different recording modality, single-neuron electrophysiology, for the reconstruction task.

## 3 STSBENCH

**Overview of dataset.** STSBench contains the activity of 2,244 neurons in the STS in response to ∼4,500 unique, 200 ms natural video clips from the Ego4D dataset, along with the stimuli and metadata associated with each neuron. We provide a complete description of the neural data files in the data repository, and we describe the tasks, recording setup, and preprocessing steps below.

**Subjects.** Neural recordings were collected in a total of eight sessions from two male Rhesus macaques (A: age 14 years, weight 11 kg, T: age 11 years, weight 10 kg). All surgical and experimental procedures were approved by the Stanford University Institutional Animal Care and Use

Committee and were in accordance with the policies and procedures of the National Institutes of Health.

**Task and recording setup.** Neural recordings were conducted in the STS while monkeys performed receptive field mapping and video fixation tasks. The location of the STS was identified using anatomical MRIs in both monkeys (**Figure 1d**) and confirmed based on the functional properties of recorded neurons. Neural data was recorded with a Neuropixels 1.0 NHP probe, a high-density extracellular electrode with 384 recording contacts [33], positioned in the STS (**Figure 1d**).

Experiment code was written in MATLAB (MathWorks, version R2020b) using the Psychophysics Toolbox [41]. Stimuli were presented on a ViewPixx LCD monitor with 1920×1080 pixels resolution and 100 Hz refresh rate (VPixx Technologies). The monkeys viewed the display from a distance of 42 cm. Eye position was monitored at 1000 Hz using monocular corneal reflection and pupil tracking with an Eyelink 1000 Plus (SR Research Ltd., Ottawa, ON, Canada). Eye-tracker calibration was performed with a five point protocol at the beginning of each recording session.

**Video fixation task.** In the video fixation task, each trial started when the monkey fixated a central spot (blue, 0.5° for monkey T; red, 1° for monkey A) for 300 ms. The fixation spot was presented offset from the center of the screen (location reported individually for each session in the dataset) on a mid-gray (33 cd/m$^2$) background. A sequence of full-screen videos (9 for monkey T, 6 for monkey A) were then displayed, each for 200 ms with no inter-stimulus interval (**Figure 1e**). The monkey received a juice reward for maintaining fixation for the duration of the trial.

The videos shown in each trial were selected from a collection of 4,533 egocentric videos from the Ego4D dataset [42]. Videos were resized to 640 x 360 to match the aspect ratio of the display and sampled at 24 frames per second such that each 200 ms video contained 5 frames. From this collection, we randomly selected 40 videos as test stimuli and used the remaining 4,493 videos as train stimuli. In each session, test stimuli were shown in a fixed proportion of trials (30% for T; 20% for A) and train stimuli were shown in the remaining trials. In each trial, stimuli to display were drawn randomly without replacement from the train or test set, and test set stimuli were cycled once all test set stimuli had been displayed. In total, 1,003-3,822 (mean 2,215) unique train videos were shown per session.

**Receptive field mapping task.** In the receptive field mapping task, the monkey fixated a central spot (white, 0.5° for T; red, 1° for A) on a mid-gray (33 cd/m$^2$) background for 300 ms to initiate a trial. A series of receptive field mapping stimuli (20 for monkey T, 13 for monkey A) were then presented at various eccentricities, each for 100 ms with no inter-stimulus interval. The stimuli in the RF mapping task were sinusoidal gratings (spatial frequency 1 cycle/°) inside a Gaussian envelope (sigma 1°) with eight different carrier orientations. The carrier wave changed phase continuously at 15 Hz. The monkey received a juice reward for maintaining fixation for the duration of the trial.

**Neural data preprocessing.** Neural data was spike-sorted using the Kilosort4 algorithm with default parameters to obtain spike times for individual neurons and waveform templates [43]. All single and multi-units identified by Kilosort4 were grouped together and are referred to as 'neurons' hereafter. The firing rate for each neuron in the video fixation task was computed over the window 40-240 ms after stimulus onset, selected based on the minimum response latency of neurons in MT/MST [44]. For each neuron, firing rates on the train and test set were z-scored using train set statistics. Firing rate on the test set was then averaged over repeated presentations of the same stimuli. We included neurons with reasonable average firing rates (>2 Hz on train set) and reliable visual responses ('reliability' > 0.5) in subsequent analyses, where 'reliability' is a bootstrapped estimate of the Pearson correlation coefficient between a neuron's firing rate computed over separate subsets of stimulus repeats. High response 'reliability' indicates that a neuron responds similarly to repeated presentations of the same stimulus and differently to presentations of different stimuli.

To identify putative cell types in the dataset, neuronal waveform templates were classified as axonal-spiking (AS), regular-spiking (RS; putative excitatory neurons), and fast-spiking (FS; putative inhibitory) based on their trough-to-peak duration [45–47]. Waveforms with initial peaks (i.e., negative trough-to-peak duration), were classified as AS. Waveforms with trough-to-peak durations between 0 and 200 μs, were classified as FS, and those with trough-to-peak durations greater than 200 μs, were classified as RS. **Figure 2a-c** shows an example waveform of a regular-spiking neuron and

example visual receptive field, along with the distribution of waveforms and receptive fields along the length of the probe in each of the eight recording sessions in the dataset. **Figure 2d-f** illustrate the temporal dynamics of neural activity around stimulus onset. Taken together, these results highlight the scale and diversity of the data in STSBENCH and its potential utility for modeling efforts.

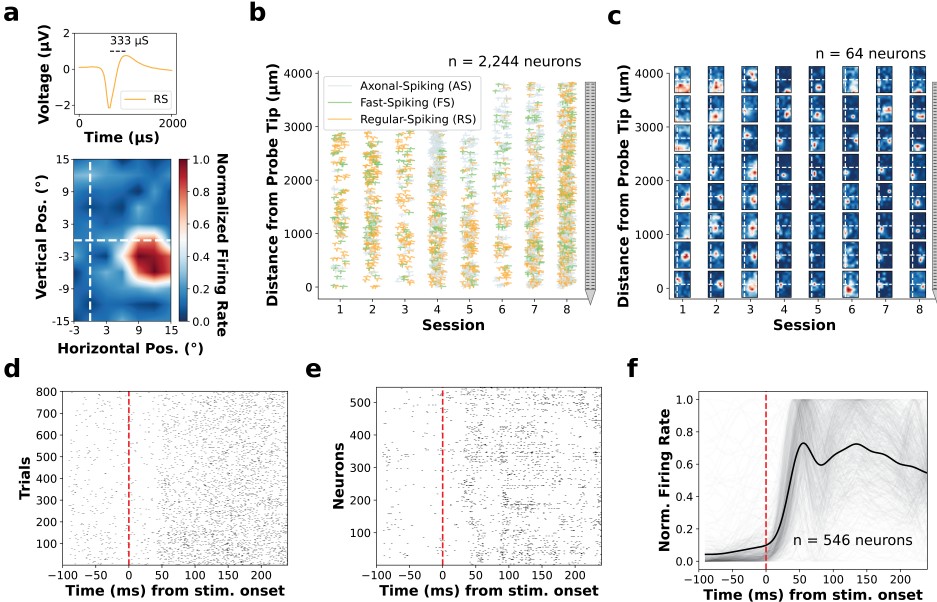

Figure 2: Overview of neural data in STSBENCH. (**a**) Waveform template for an example neuron, and corresponding visual receptive field mapped with drifting gratings. In the receptive field plot, white dashed lines indicate the horizontal and vertical meridians, and units are degrees of visual angle. (**b**) Waveform templates for all neurons plotted along the length of the probe in each recording session. (**c**) Receptive fields for eight example neurons per session at different positions along the probe. Schematic of Neuropixels probe is shown on the right in (b-c) (not to scale). (**d-e**) Raster plots display activity of a single neuron across all trials in a session (d) and activity of the population of neurons in a single trial (e) aligned to stimulus onset. (**f**) Peristimulus time histograms (PSTHs) are plotted for single neurons (light grey) and averaged over all neurons in a session (black).

## 4 Encoding and reconstruction models

### 4.1 Encoding models

The encoding models evaluated here consist of a feature extractor that embeds an input video followed by a readout layer that predicts a neuron's firing rate from that embedding (**Figure 3a**). The parameters of the feature extractor are shared across neurons while the parameters of the readout are unique to each neuron. The input to the model is a $150 \times 150$ crop of the $360 \times 640$ video shown to the monkey, selected to cover the receptive fields of most neurons in the dataset. The output of the model is a scalar that represents a prediction of the z-scored firing rate of a single neuron. We provide an overview of the encoding models below and more detail in Appendix A.

**Feature extractors.** We compare a suite of convolutional neural network (CNN) feature extractors with weights that were either hand-tuned, pre-trained, or trained end-to-end. The hand-tuned 3D Gabor model and pretrained ResNet models were proposed in previous studies as models of MT/MST [28, 26, 48, 49]. The 3D CNN models that are trained end-to-end have 1-7 layers, 32 channels per layer, and batch normalization and ReLU nonlinearities between layers. As in previous studies [17, 34, 24], we compare the effect of using layers at different depths in the pretrained networks and the interaction between layer and input size. Testing different input sizes is important for pretrained models where filter weights are fixed but not for models trained end-to-end that can learn the appropriate filters at different resolutions.

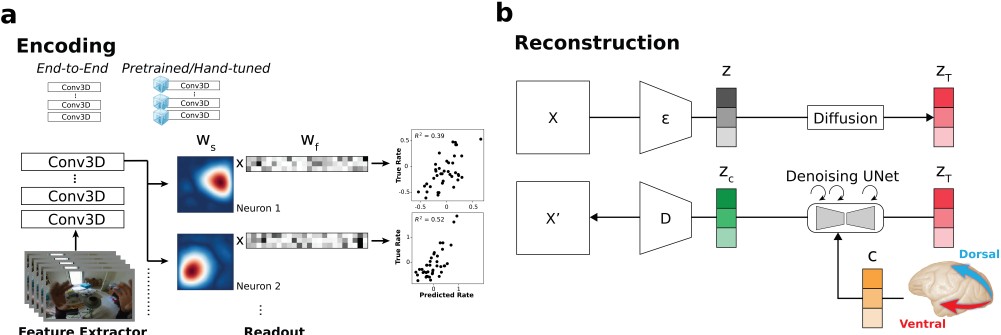

Figure 3: Overview of encoding and reconstruction models. **(a)** Encoding models consist of a feature extractor that embeds the input video using 3D convolutions and a factorized readout layer that predicts a neuron's firing rate from that embedding using spatial ($W_s$) and feature ($W_f$) weights. The feature extractor is either hand-tuned/pretrained and frozen or trained end-to-end. Scatter plot shows predicted firing rate vs true firing rate on the test set for two example neurons predicted with the 3D ResNet-Kinetics, Layer 2 model. **(b)** Reconstruction models consist of a VQ-VAE ($\varepsilon$ and $D$) that maps an image ($X$) to a latent representation ($z$), and a latent diffusion model that generates an image conditioned on neural activity ($c$).

**Readouts.**    For each neuron, the readout maps a video embedding $\mathbf{x} \in \mathbb{R}^{c \times d \times w \times h}$ (**c**hannels, **d**epth, **w**idth, **h**eight) to a scalar that represents the neuron's response. The mapping is affine with weights $\mathbf{w} \in \mathbb{R}^{c \times d \times w \times h}$. As in previous studies [50–52, 24], we factorize $\mathbf{w}$ into a set of feature weights $w_{i,j}^f$ and spatial weights $w_{k,l}^s$ such that $w_{i,j,k,l} = w_{i,j}^f w_{k,l}^s$.

## 4.2   Reconstruction models

The primary reconstruction model evaluated here is a conditional latent diffusion model that is trained to generate an image from neural activity (**Figure 3b**). The model consists of a vector-quantized variational autoencoder (VQ-VAE) and denoising U-Net. The input to the reconstruction model is the vector of firing rates of all MT/MST neurons in STSBENCH or all V4 neurons in TVSD, and the output is a $256 \times 256$ color image. In STSBENCH, this is a resized version of the $150 \times 150$ encoding model crop described above, and in TVSD this is a resized version of an uncropped image.

Our reconstruction model is based on Stable Diffusion [53], a text-conditional latent diffusion model. In Stable Diffusion, cross-attention layers in the U-Net backbone incorporate information from a text embedding (e.g., from CLIP [54]) during generation. In our reconstruction model, cross-attention layers incorporate information from a neural activity vector that contains the firing rates of neurons in response to a particular stimulus. Concretely, we replace the text embedding $\mathbf{e} \in \mathbb{R}^{B \times T \times D}$ (**B**atch size, **T**okens, **D**imension of text embedding) in Stable Diffusion with neural activity $\mathbf{c} \in \mathbb{R}^{B \times 1 \times N}$ (**B**atch size, 1, **N**eurons). This approach worked out-of-the-box with the same hyperparameters used to train text-conditional latent diffusion models. Our implementation, hyperparameter choices, and training details follow the text-conditional latent diffusion model in the StableDiffusion-PyTorch repository [55]. Both the VQ-VAE and diffusion model are trained from scratch on the data in STSBENCH or TVSD. We document all hyperparameters and training settings in the associated code.

To quantify model performance, we report the peak-signal to noise ratio (PSNR) which is inversely related to mean-squared error and quantifies pixel-level similarity between two images, and Learned Perceptual Image Patch Similarity (LPIPS) which captures perceptual similarity by comparing deep feature representations in a pretrained network (here, AlexNet) [56]. We compare the diffusion model to two null models. The 'shuffled' null model compares each test set image to other images drawn from the test set to estimate the performance of an unconditional generative model. The 'mean' null model compares each test set image to the mean image to quantify the optimal PSNR of an unconditional generative model.

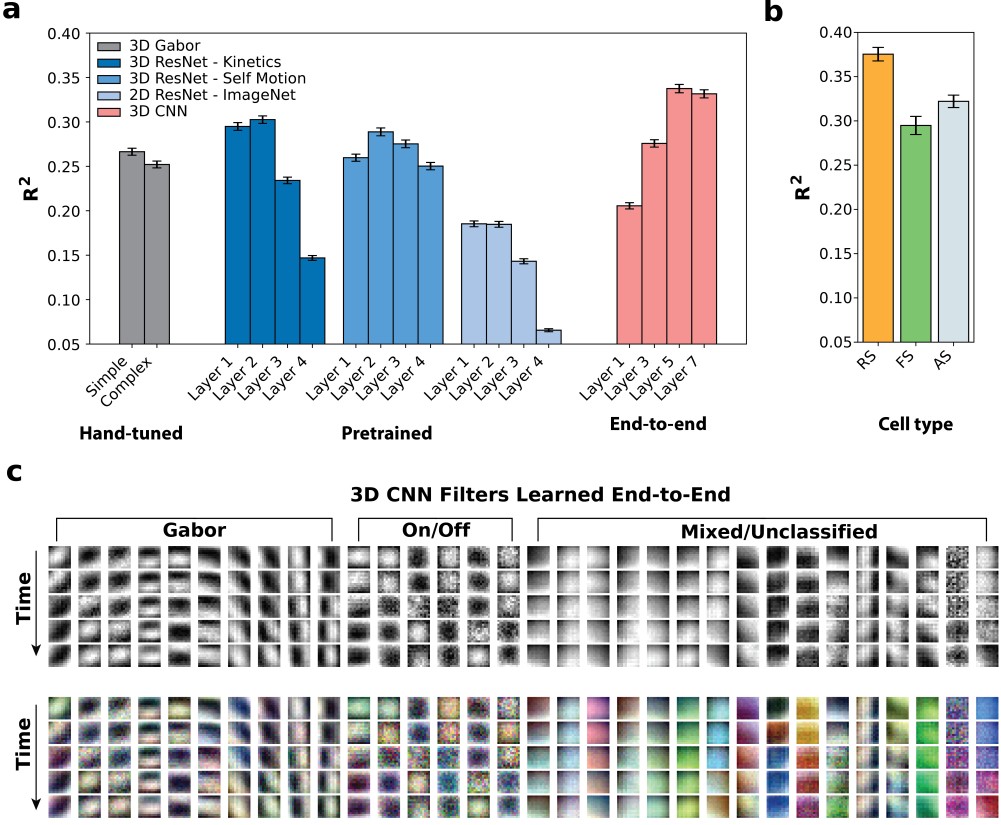

Figure 4: Encoding model results. (**a**) Performance ($R^2$, mean $\pm$ s.e.m.) of each encoding model on the test set, separated by training scheme (hand-tuned, pretrained, end-to-end) and layer used to readout neural activity. (**b**) Performance ($R^2$, mean $\pm$ s.e.m.) of the 3D CNN-5 encoding model trained end-to-end, separated by functional cell type. (**c**) 3D convolutional filters in the 3D CNN-1 encoding model trained end-to-end, with (top) and without (bottom) averaging over color channels. Filters were manually grouped based on shared features, such as similarity to drifting Gabors.

## 5 Results

### 5.1 Encoding

We first tested whether we could predict the activity of individual neurons in STSBENCH from the video shown to the monkey. We tested encoding models with feature extractors that were either hand-tuned, pretrained, or trained end-to-end. Our motivation for developing and benchmarking such video-computable encoding models with STSBench was twofold. First, by testing encoding models trained end-to-end against pretrained or hand-tuned baselines, one might identify gaps in current conceptual models and theories of dorsal stream processing. Second, by examining the features learned by encoding models trained end-to-end, one might gain an intuitive understanding of the features of the visual world that the dorsal stream encodes without imposing restrictive inductive biases. We found that 3D CNN models trained end-to-end outperformed a suite of baseline models proposed in previous studies (**Figure 4a**), including a model that uses features from a 3D ResNet pretrained on a self motion estimation task [28] and a model that uses a pyramid of hand-tuned 3D Gabor filters [26]. This result suggests that while models based on hand-tuned Gabor filter banks [26] or filters optimized for self-motion estimation [28] provide solid baselines, they may not capture responses in MT/MST as well as models trained end-to-end.

We next examined the dependence of model performance on overall network depth. For the 3D CNN models trained end-to-end, we found that performance increased with increasing depth and plateaued after five layers, highlighting the importance of deep nonlinear computations for predicting

responses in MT/MST (**Figure 4a**). For the 3D ResNet model pretrained on Kinetics, features in earlier layers were better predictors of neural activity than features in later layers, suggesting that neurons in MT/MST are more selective to lower level visual representations than features necessary for classifying actions (**Figure 4a**). Additionally, encoding model performance depended on putative cell type, with better performance for regular-spiking (putative excitatory) neurons than fast-spiking (putative inhibitory) neurons (**Figure 4b**; two-sample t-test, $p < 10^{-5}$). This observation is consistent with evidence of weaker selectivity among cortical interneurons compared to pyramidal neurons [57]. We analyze the dependence of these results on input size and report additional quantitative results in Appendix B.

Finally, we examined the filters learned by the 1-layer 3D CNN model trained end-to-end to understand the features that MT/MST neurons encode. Interestingly, 10 out of the 32 filters learned by the 1-layer 3D CNN were similar to drifting Gabor patches with different orientations (**Figure 4c**), which aligns with our current conceptual understanding of MT [26]. The remaining filters showed a mixture of more complex properties such as 'on-off' responses and rotations. We also observed chromatic features in many of the filters which are not present in the textbook model of MT/MST [although see 58]. We leave a more thorough investigation of the nonlinear features and circuits learned by the deeper and more performant 5-layer 3D CNN to future studies.

## 5.2 Reconstruction

We next asked whether we could reconstruct the visual stimulus shown to the monkey from neural activity in STSBENCH. In early visual structures, such as the retina, that encode all features of a visual scene, reconstructions should be able to capture all features of an input stimulus veridically. In contrast, in higher visual areas where visual representations are factorized, reconstructions should only capture the features encoded by neurons in that area. Thus, if our conceptual model of the dorsal and ventral visual streams is correct, then reconstructions from the dorsal stream should capture primarily motion statistics and low-spatial frequency contours ('where') whereas reconstructions from the ventral stream should capture high-spatial frequency details, including color, texture, form, and object identity ('what') [59, 10]. To assess whether information about motion statistics and low-spatial frequency contours is encoded by neurons in STSBENCH, we trained an image reconstruction model to reconstruct the first frame of each video from neural activity (**Figure 5a**), a decoder to predict the average motion direction of each video (Appendix D), and a grayscale video reconstruction model to reconstruct all frames of each video (Appendix E). We also trained the image reconstruction model on neural data from the mid-level ventral stream area V4 [9] from TVSD for reference (**Figure 5b**).

We found that image reconstructions from neural activity in the dorsal stream qualitatively captured low-spatial frequency luminance contours, such as the edges of computers, shelves and tables (**Figure 5a**). High-spatial frequency and object identity information, such as the water bottles on the shelf and apple logo on the computer, were notably absent (**Figure 5a**). Quantitatively, the conditional diffusion model performed substantially better than 'shuffled' and 'mean' null models on both PSNR and LPIPS (**Table 1**), indicating that the model utilizes neural activity to condition image generation. Moreover, results from the motion direction decoder (Appendix D) and video reconstruction model (Appendix E) suggest motion information is encoded by neurons in STSBENCH.

In the ventral stream, reconstructions captured detailed information about form, object identity and color, such as the shape of a dough ball, presence of a monkey, and color of a mango (**Figure 5b**). Quantitatively, the conditional diffusion model performed substantially better than the 'shuffled' and 'mean' null models on LPIPS but not PSNR (**Table 1**).

Table 1: Reconstruction results for dorsal and ventral streams. Best value for each metric in each column is bolded (max for PSNR, min for LPIPS).

| Model | Dorsal Stream (MT/MST) | | Ventral Stream (V4) | |
|---|---|---|---|---|
| | PSNR | LPIPS | PSNR | LPIPS |
| Mean | 11.335 | 0.873 | **12.192** | 0.912 |
| Shuffled | 9.848 | 0.751 | 9.333 | 0.690 |
| Diffusion | **14.161** | **0.668** | 10.631 | **0.589** |

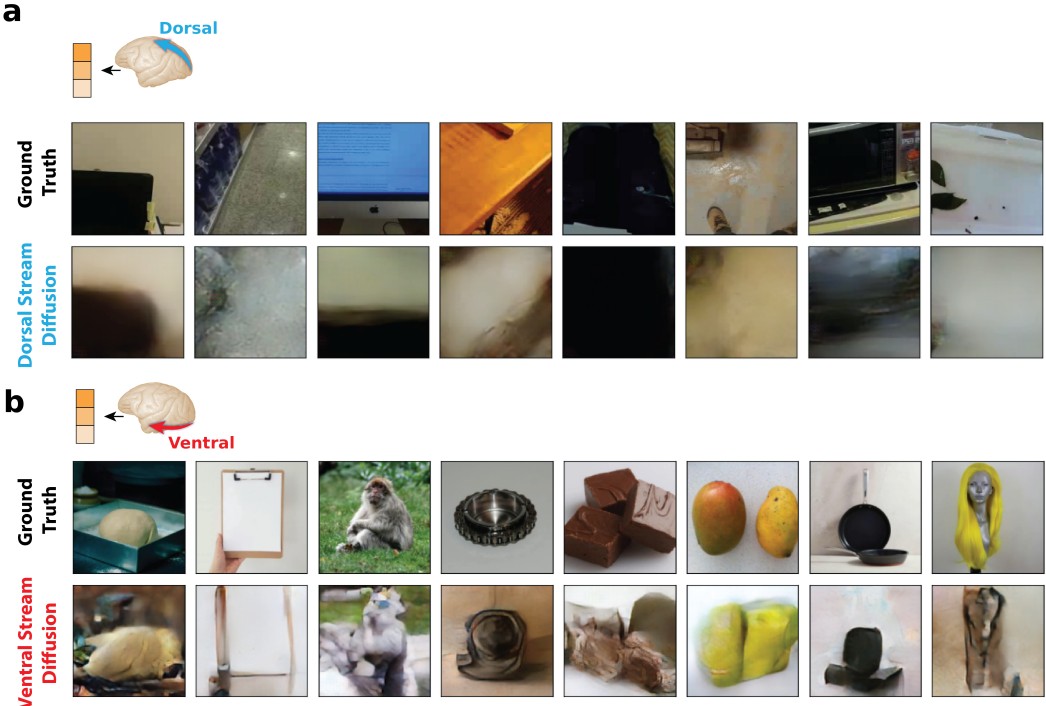

Figure 5: Dorsal (MT/MST) and ventral (V4) stream reconstruction results. Example test set images (top row), and their corresponding reconstructions from neural activity (bottom row). (**a**) The dorsal stream reconstructions primarily capture low spatial frequency components of the scene, a characteristic feature of dorsal stream representations. (**b**) The ventral stream reconstructions capture form, color, and object identity, key characteristics of ventral stream representations.

Taken together, the dorsal stream reconstruction results illustrate that the features of visual stimuli known to be represented by the dorsal stream, such as motion and luminance contrast, can be recovered from neuronal activity in STSBENCH. The ventral stream reconstruction results validate our modeling approach and provide a qualitative reference, but direct comparisons between the datasets cannot be made due to differences in neuron counts, preprocessing techniques (spike-sorted in STSBENCH vs threshold-crossing in TVSD), and stimuli (natural videos in STSBENCH vs natural images in TVSD). See Appendix C for additional results with linear and CNN baselines.

## 6    Conclusion

Here, we presented STSBENCH, a large-scale dataset of neuronal recordings from the STS collected while monkeys viewed natural videos. We showed that STSBENCH can be used for training encoding models of MT/MST and reconstructing visual stimuli from neural activity. Although the encoding and reconstruction models used here are simple extensions of standard approaches in machine learning, our results highlight the power of leveraging these techniques to better understand neural circuits in the brain and stress test theories of visual processing. Our encoding results demonstrate that simple 3D CNNs trained end-to-end outperform other baseline models of MT/MST, underscoring the potential of STSBENCH to refine models of dorsal stream visual processing. Our reconstruction results highlight the utility of STSBENCH for studying the features of the visual world represented by populations of neurons in the dorsal stream.

The dataset and baselines presented here are an important first step towards developing a more comprehensive understanding of dorsal stream visual areas and pave the way for future studies with STSBENCH. Though we aimed to incorporate a representative set of encoding models, the set of models we tested was not exhaustive. For example, there are other variants of the Gabor filter bank

model [26] and different readout mechanisms for predicting activity from pretrained networks that could be examined [52]. By providing STSBENCH to the community, we hope to enable more complete and comprehensive evaluations of models of MT/MST in future studies. Another promising direction for future work is to apply interpretability methods [e.g. 60] to understand the nonlinear computations in the 5-layer 3D CNN model of MT/MST that outperformed other baselines. This effort could give rise to a more nuanced and complete mechanistic understanding of MT/MST.

# 7   Limitations

There are a number of important limitations of the dataset and the analyses presented here. First, although the recordings in STSBench primarily encompass neurons in MT/MST, a few of the superficial neurons in sessions 1-3 may be from the adjacent higher level dorsal stream area 7a and a few of the superficial neurons in sessions 4-5 may be from adjacent white matter. Second, although we applied a standard automated preprocessing method for assigning spikes in the voltage trace to individual neurons (Kilosort 4.0), this preprocessing pipeline can overcount neurons ("split" errors) or incorrectly group spikes from two real neurons into a single neuron ("merge" errors). Third, our use of RS and FS waveform characteristics to distinguish between putative excitatory and inhibitory neurons, though common in the extracellular neurophysiology, likely underestimates the diversity of cell types.

# 8   Societal Impact

This work contributes to our understanding of the fundamental neural mechanisms underlying visual processing in the primate brain. More broadly, we hope that developing a deeper understanding of biological vision algorithms will aid efforts to improve the efficiency, robustness, and interpretability of computer vision models.

# 9   Acknowledgements

We are grateful to Katrin Franke, Nikos Karantzas, and Andreas Tolias for fruitful discussions on this work. This work was supported by NIH EY014924, NS116623 and a Ben Barres Professorship to T.M., and NSF GRFP 2146755 and a Stanford Bio-X Graduate Student Fellowship to E.T.

# 10   Data and Code Availability

The STSBENCH dataset is available on Kaggle (`https://www.kaggle.com/datasets/ethantrepka1/stsbench/`). The associated code is available on GitHub (`https://github.com/et22/stsbench`).

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

# A  Encoding model methods

## A.1  Feature extractors

*End-to-end - 3D CNNs.* We evaluate a family of simple 3D convolutional neural network (*3D CNN-X*) models with 1, 3, 5, or 7 layers and 32 channels per layer. Each convolutional layer is followed by batch normalization and a ReLU nonlinearity. The first convolutional layer uses kernels of size $5 \times 11 \times 11$ (depth $\times$ width $\times$ height), and all subsequent layers use $3 \times 3 \times 3$ kernels. The first three layers use a spatial stride of 2, while subsequent layers use a stride of 1. For the 1-layer model, we add an average pooling layer with a stride of 4 to match the size of output feature map of deeper models.

*Pretrained - 2D and 3D ResNets.* We evaluate three pretrained ResNet models: an 18-layer 3D ResNet pretrained on the Kinetics action recognition task (*3D ResNet-Kinetics*, weights from torchvision) [49, 61], a 6-layer 3D ResNet pretrained on a self motion estimation task (*3D ResNet-Self Motion*) [28], and an 18-layer 2D ResNet pretrained on ImageNet (*2D ResNet-ImageNet*, weights from torchvision) [48, 61]. The 2D ResNet processes only the first frame of each video. The layer numbering conventions for the *3D ResNet-Self Motion* model follow Mineault et al. [28], while the conventions for the other ResNet models follow the PyTorch implementations.

*Hand-tuned - 3D Gabor model.* We evaluate a classic spatiotemporal receptive field model, in which 3D Gabor filters with different drift directions, phases, and spatial frequencies are convolved with the input video to generate a feature map [26, 28]. Our implementation follows Mineault et al. [28], and includes both a *simple* cell layer where a ReLU nonlinearity is applied to the feature map before the readout, and a *complex* cell layer where the input to the readout is the norm of pairs of filters with identical parameters but 90° phase offsets [26].

## A.2  Training and evaluation details

The Adam optimizer was used to train all encoding models with a learning rate of 0.001. Training data was divided into train and validation sets with a 90/10 split. We optimized a loss function consisting of three terms: a mean squared error term for predicting firing rates, a Laplacian regularization term that encourages spatial smoothness in the receptive field weights [51, 24], and an $L_2$ penalty to prevent overfitting. The regularization strengths $\lambda_1$ and $\lambda_2$ of the Laplacian and $L_2$ terms, respectively, were selected for each model separately via a grid search to maximize performance on the validation set. This grid search required ∼1-10 hours on a single L40S GPU depending on the feature extractor used. $R^2$ between the predicted and true neuronal response on the held-out test set is reported in all tables and figures.

# B   Encoding model results

Table 2: Full encoding model results. $R^2$ results on the test set are reported as mean $\pm$ s.e.m.

| Training Scheme | Model Name | Layer | Input Size (px) | $R^2$ |
|---|---|---|---|---|
| Hand-tuned | 3D Gabor | Simple | 112 | $0.244 \pm 0.004$ |
| | | | 224 | $0.266 \pm 0.004$ |
| | | Complex | 112 | $0.232 \pm 0.004$ |
| | | | 224 | $0.252 \pm 0.004$ |
| Pretrained | 2D ResNet - ImageNet | Layer 1 | 112 | $0.185 \pm 0.003$ |
| | | | 224 | $0.181 \pm 0.003$ |
| | | Layer 2 | 112 | $0.185 \pm 0.003$ |
| | | | 224 | $0.182 \pm 0.003$ |
| | | Layer 3 | 112 | $0.143 \pm 0.003$ |
| | | | 224 | $0.165 \pm 0.003$ |
| | | Layer 4 | 112 | $0.066 \pm 0.002$ |
| | | | 224 | $0.090 \pm 0.002$ |
| | 3D ResNet - Kinetics | Layer 1 | 112 | $0.301 \pm 0.004$ |
| | | | 224 | $0.295 \pm 0.004$ |
| | | Layer 2 | 112 | $0.293 \pm 0.004$ |
| | | | 224 | $0.303 \pm 0.004$ |
| | | Layer 3 | 112 | $0.206 \pm 0.003$ |
| | | | 224 | $0.234 \pm 0.004$ |
| | | Layer 4 | 112 | $0.082 \pm 0.002$ |
| | | | 224 | $0.147 \pm 0.003$ |
| | 3D ResNet - Self Motion | Layer 1 | 112 | $0.260 \pm 0.004$ |
| | | | 224 | $0.249 \pm 0.004$ |
| | | Layer 2 | 112 | $0.289 \pm 0.004$ |
| | | | 224 | $0.259 \pm 0.004$ |
| | | Layer 3 | 112 | $0.275 \pm 0.004$ |
| | | | 224 | $0.248 \pm 0.004$ |
| | | Layer 4 | 112 | $0.250 \pm 0.004$ |
| | | | 224 | $0.222 \pm 0.004$ |
| End-to-end | 3D CNN-1 | Layer 1 | 64 | $0.206 \pm 0.004$ |
| | 3D CNN-3 | Layer 3 | 64 | $0.276 \pm 0.004$ |
| | 3D CNN-5 | Layer 5 | 64 | $\mathbf{0.338 \pm 0.005}$ |
| | 3D CNN-7 | Layer 7 | 64 | $0.332 \pm 0.005$ |

## C    Reconstruction model baselines and results

We also evaluated linear and CNN decoders as minimal baselines to test the feasibility of directly reconstructing images from neural activity. The linear decoder consists of a single fully connected layer that maps the neural activity vector to a flattened image tensor. The CNN decoder uses a fully connected layer to project the neural activity vector into a latent representation of size $128 \times 8 \times 8$. This latent code is then upsampled through a sequence of transposed convolution layers with batch normalization and ReLU nonlinearities to the final image resolution. For both the linear and CNN decoder, the output is passed through a scaled sigmoid activation $(2 \cdot \text{sigmoid}(x) - 1)$ to generate a $32 \times 32$ image with pixel values in the $[-1, 1]$ range. Linear and CNN baselines were trained to minimize mean-squared error between the predicted and true image using the Adam optimizer with a learning rate of 0.001. The qualitative and quantitative reconstruction results for these models, compared to the diffusion model and other controls, are illustrated below.

Training the linear and CNN reconstruction models required $< 1$ hour on one L40S GPU. Training the diffusion models was more compute intensive. Pretraining the VQ-VQE model required $\sim$3 hours on one L40S GPU for each dataset and training the latent diffusion model required $\sim$24 hours on one L40S GPU for each dataset.

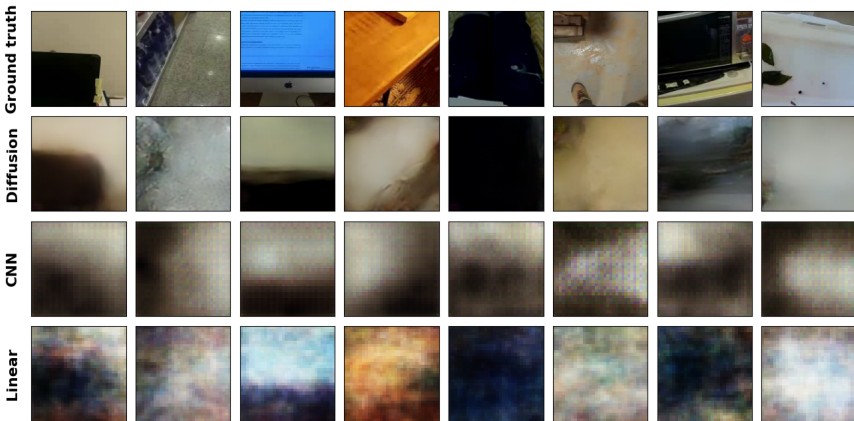

Figure 6: Dorsal stream (STS) reconstruction results with additional baselines. Example test set images reconstructed from neural activity with a conditional diffusion model, CNN, or linear model. Ground truth test image is shown in the top row.

Table 3: Dorsal stream (STS) reconstruction results with baselines. Best value for each metric is bolded (max for PSNR, min for LPIPS).

| Model | PSNR | LPIPS |
|---|---|---|
| Mean | 11.335 | 0.873 |
| Shuffled | 9.848 | 0.751 |
| Linear | **14.818** | 0.820 |
| CNN Decoder | 11.463 | 0.838 |
| Diffusion | 14.161 | **0.668** |

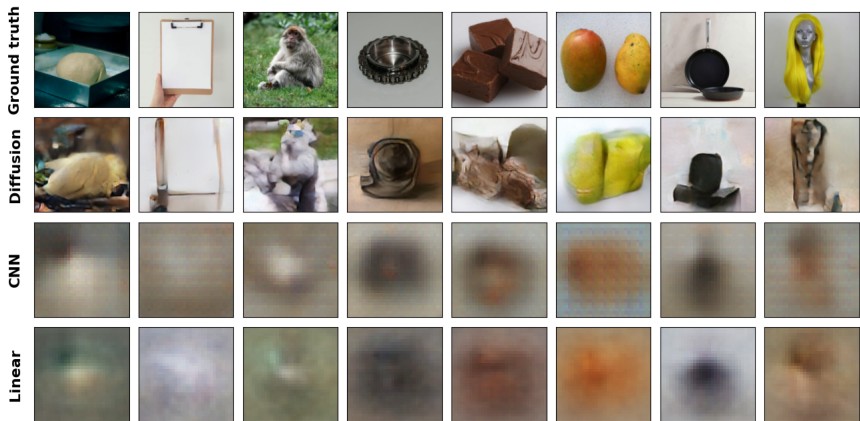

Figure 7: Ventral stream (V4) reconstruction results with baselines. Example test set images reconstructed from neural activity with a conditional diffusion model, CNN, or linear model. Ground truth test image is shown in the top row.

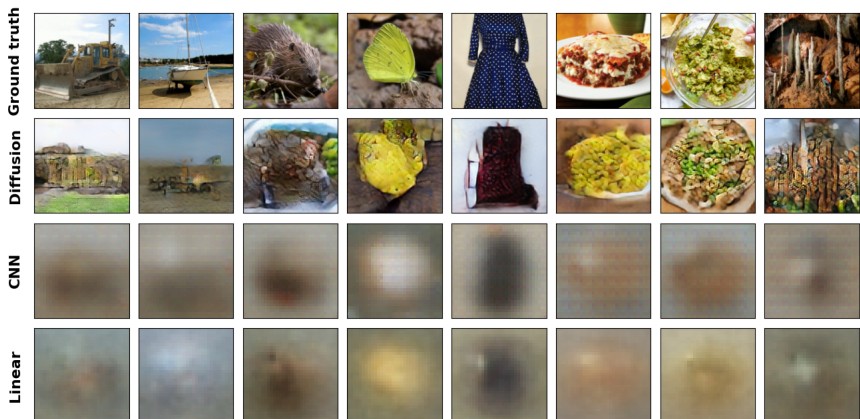

Figure 8: Additional ventral stream (V4) reconstruction results with baselines.

Table 4: Ventral stream reconstruction results with baselines. Best value for each metric is bolded (max for PSNR, min for LPIPS).

| Model | PSNR | LPIPS |
|---|---|---|
| Mean | 12.192 | 0.912 |
| Shuffled | 9.333 | 0.690 |
| Linear | **13.313** | 0.920 |
| CNN Decoder | 12.821 | 0.954 |
| Diffusion | 10.631 | **0.589** |

## D   Decoding optic flow

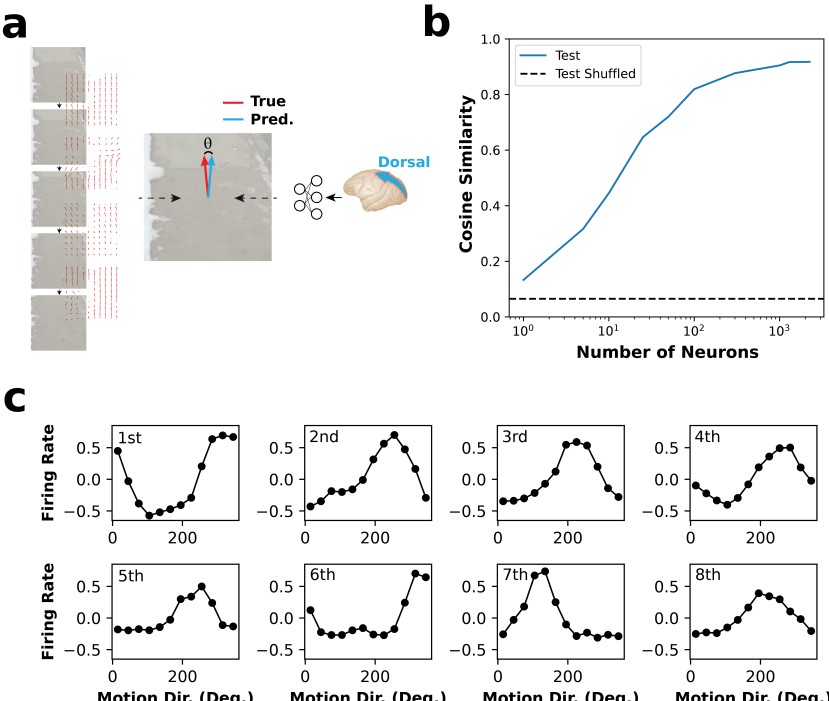

Figure 9: Decoding optic flow from neural activity in STSBENCH. (**a**) Diagram of optic flow calculation and decoding for an example test set video. Dense optical flow is computed using the Farneback method across adjacent frames, then averaged over time and space to assign a single motion direction vector to each video. A linear regression model is trained to predict the motion direction vector from neural activity in STSBENCH. (**b**) Decoding performance (cosine similarity) on the held-out test set as a function of the number of neurons used for decoding. The horizontal line denotes the performance of a target-shuffled control. (**c**) Feature importance analysis. The normalized firing rates of the top eight features (neurons) in the decoder are plotted as functions of motion direction.

To assess whether neural activity in STSBENCH encodes motion information, we trained a regression model to predict the direction of motion in each video from neural activity. Motion direction labels were computed by applying the Farneback algorithm [62] to estimate dense optical flow between adjacent video frames. The resulting flow vectors were averaged over both space and time, then normalized to obtain a unit vector representing the dominant motion direction for each video (**Figure 9a**).

The input to the decoder was the vector of firing rates of all neurons in STSBENCH associated with a particular video. The output of the decoder was a prediction of the $x$ and $y$ values of the normalized motion direction vector (**Figure 9a**). The RidgeCV ridge regression model in scikit-learn was used as the decoder [63]. The model was trained on the STSBENCH training set, then evaluated on the held-out test set by measuring the cosine similarity between the predicted and true motion direction vectors. As a control, we trained the same model with randomly shuffled targets and averaged results over five shuffles.

The model accurately decoded motion direction, achieving a mean cosine similarity of $0.923$ on the test set compared to $0.051$ for the shuffled control (**Figure 9b**), indicating that motion direction can be robustly decoded from neural activity.

To examine the dependence of decoding performance on the number of neurons used as predictors, we trained the decoder with subsets of the total population. We observed a linear relationship between the logarithm of the number of neurons and cosine similarity that began to saturate at $\sim 100$ neurons (**Figure 9b**).

To identify the neurons that most strongly influenced the decoder output, we computed a feature importance score for each neuron, defined as the sum of the absolute values of its weights for predicting the $x$ and $y$ components of the motion direction vector. We then examined the firing rates of the eight most informative neurons as a function of video motion direction. The neurons exhibited clear Gaussian-like tuning to motion direction (**Figure 9c**), underscoring the potential utility of STSBENCH for studying how the dorsal stream processes motion.

# E  Grayscale video reconstruction

To further assess the presence of motion information in neural activity in STSBENCH, we trained a model to reconstruct five-frame grayscale videos conditioned on neural activity. The VQ-VAE and diffusion model architecture and training procedure were identical to the image reconstruction models, except that the first and last layers of the VQ-VAE were adapted to process inputs and outputs of shape $[5, W, H]$ rather than $[3, W, H]$, replacing the color channels with a time dimension. For evaluation, we report the cosine similarity between the average motion direction in the ground truth video and its reconstruction, computed as in Appendix D.

The reconstruction model achieved a cosine similarity of 0.438 between the average motion direction of the reconstructed and ground truth videos, substantially higher than chance (0 if motion direction is uniformly distributed; 0.0485 with the circular mean of the training set). This is well below the cosine similarity of 0.923 obtained in Appendix D when decoding motion direction directly from neural activity, suggesting that there is considerable room for improving upon these results. Example reconstructions are illustrated in **Figure 10** and corresponding videos are included in the code repository.

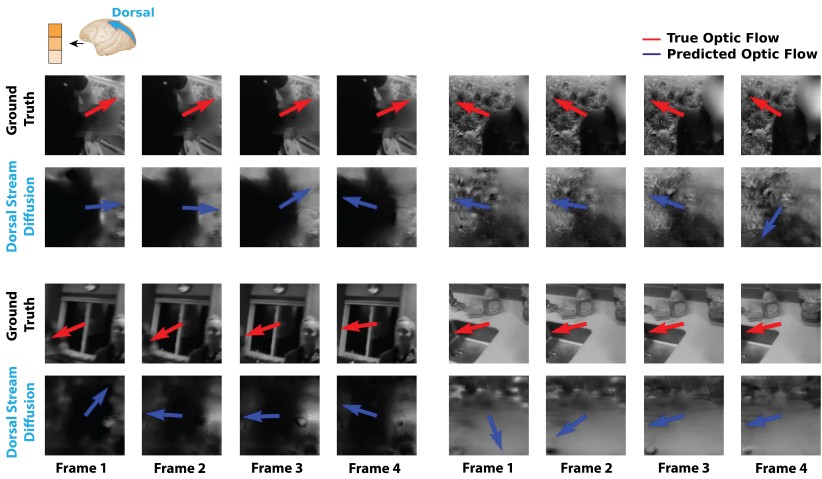

Figure 10: Reconstructing grayscale videos from neural activity in STSBENCH. Four example ground truth videos and corresponding reconstructions are displayed. Arrows denote the mean optic flow direction for adjacent frames computed using the Farneback method in either the ground truth video (red) or reconstructed video (blue).

