# OpenReview forum: "STSBench: A Large-Scale Dataset for Modeling Neuronal Activity in the Dorsal Stream of Primate Visual Cortex"
_NeurIPS.cc/2025/Datasets_and_Benchmarks_Track — NeurIPS 2025 Datasets and Benchmarks Track poster_

### Official Review · Reviewer_2hUT · 2025-06-26

**Rating:** 5
**Confidence:** 4

**Summary:**

The authors present a new dataset of neural recordings using neuropixel electrodes in the macaque dorsal stream, while the animals were viewing natural scene videos. This dataset contains recordings of more neurons than any previous dorsal stream dataset I know of. The authors share this dataset openly. They also run some preliminary encoding and image reconstruction analyses. In encoding, 3D CNNs outperformed hand-crafed gabor features, in line with many findings in recent years where optimized features outperform hand-crafted features. In image reconstruction, they attempt to reconstruct the image that monkeys were seeing, based on the monkey's recorded brain activities. They find that the reconstructions from dorsal stream have less details about objects than reconstructions from another dataset of ventral stream activities, in line with neuroscientific knowledge.

**Additional Feedback:**

I have never attended Neurips, so I do not know how common it is to have neuroscience papers. This is a strong neuroscience contribution, but does provide direct advancements in AI. In my opinion, that is ok, since neuroscience and AI have often synergized.

**Dataset Code Accessibility:**

Yes

**Dataset Code Comments:**

The python snippets provided on this page did not work for me (ValueError: Invalid dataset handle: datasets/ethantrepka1/stsbench/versions/1).

However, the dataset can be downloaded manually from kaggle at the link provided in the paper.

**Ethical Comments:**

I did not find an ethics declaration for the monkey experiment. Maybe I missed it. It would be good to include mention that the animal experiment was accepted by an ethics board.

**Ethical Considerations:**

No, there are no or only very minor ethics concerns

**Final Justification:**

This is a strong paper which will contribute to the field of NeuroAI. The authors engaged well with the reviewer comments. I confirm my scaore of 5 (Accept),

**Limitations Weaknesses:**

I do not see glaring weaknesses. Of course, the encoding and image reconstruction experiments are here mostly to demonstrate things that can be done with the dataset and do not have the depth and control of a paper attempting to make a novel neuroscientific claim. But I think this is understandable, given that the main focus of this paper is on the dataset itself.

**Strengths Contributions:**

Overall, the main contribution of this submission is this large, new dataset, which will undoubtedly be useful in neuroscience. As the authors mention, computational neuroscience has greatly benefited from large scale neural datasets of brain activities in the ventral visual stream. To the best of my knowledge, I agree with the authors that large scale datasets of activities in the macaque dorsal stream during natural video viewing does not exist, and will be highly valuable.

I am no expert in monkey data acquisition, so do not feel competent in assessing data quality.

The encoding and image reconstruction methodologies are sound.

---

> ### Author Rebuttal · Authors · 2025-07-30
>
> We thank the reviewer for their thoughtful evaluation of our work. We have made the following revisions to address the reviewer’s comments.
>
> **Comment 1:** *I do not see glaring weaknesses. Of course, the encoding and image reconstruction experiments are here mostly to demonstrate things that can be done with the dataset and do not have the depth and control of a paper attempting to make a novel neuroscientific claim. But I think this is understandable, given that the main focus of this paper is on the dataset itself.*
>
> **Response 1:** We agree that the primary goal of the paper is to introduce the dataset and showcase its potential applications, rather than to make novel neuroscientific claims. To clarify this for readers, we have revised our conclusions to emphasize the potential utility of the dataset for refining models of MT/MST and to avoid implying that our current experiments are sufficient to do so. Lines 269-273 now read: *"Our encoding results demonstrate that simple 3D CNNs trained end-to-end outperform other baseline models of MT/MST, underscoring the potential of STSBench to refine models of dorsal stream visual processing. Our reconstruction results highlight the utility of STSBench for studying the features of the visual world represented by populations of neurons in the dorsal stream.”*
>
> **Comment 2:** *The python snippets provided on this page did not work for me (ValueError: Invalid dataset handle: datasets/ethantrepka1/stsbench/versions/1). However, the dataset can be downloaded manually from kaggle at the link provided in the paper.*
>
> **Response 2:** We thank the reviewer for noting this. This is likely because the Kaggle dataset is currently private with link sharing enabled, so it can be accessed with the link but not the dataset handle. The visibility of the Kaggle dataset will be changed to public upon acceptance.
>
>
> **Comment 3:** *I did not find an ethics declaration for the monkey experiment. Maybe I missed it. It would be good to include mention that the animal experiment was accepted by an ethics board.*
>
> **Response 3:** We thank the reviewer for this important point. We have now moved the ethics statement from Appendix A to the main text to ensure it is not overlooked. The revised ethics statement reads:  *“All surgical and experimental procedures were approved by the Anonymous Institutional Animal Care and Use Committee and were in accordance with the policies and procedures of the National Institutes of Health.”*
>
> **Additional Notes:** In response to a comment from reviewer "L7sH", we have added motion decoding and video reconstruction results to the appendix. The text of the new appendices can be found at the bottom of our response to reviewer "L7sH".

---

> > ### Comment · Reviewer_2hUT · 2025-08-04
> >
> > I thank the authors for engaging well with the review process. All my points are addressed and I am confident in my rating of 5 (Accept).

---

### Official Review · Reviewer_L7sH · 2025-07-01

**Rating:** 5
**Confidence:** 4

**Summary:**

The paper introduces STSBench, a large-scale single-unit dataset from ~2.2 k MT/MST neurons recorded with Neuropixels while two macaques viewed ~4.5 k 200 ms natural video clips. It is positioned as the first dorsal-stream counterpart to well-known ventral-stream resources. The authors benchmark a broad panel of spatiotemporal encoding models, and train conditional latent-diffusion decoders to reconstruct frames from neural activity, contrasting dorsal vs. ventral representations.

**Additional Feedback:**

Lines 221-222: the statement that results “suggest limitations in our current understanding of MT/MST *mechanistically* as a Gabor filter bank and *conceptually* as a region optimized for self-motion estimation” is intriguing but brief. Could you elaborate on that?

**Dataset Code Accessibility:**

Yes

**Dataset Code Comments:**

Code is accessible in the supplementary materials, with a well-documented README for downloading and preprocessing the dataset and reproducing paper results. Dataset is hosted on Kaggle.

**Ethical Considerations:**

No, there are no or only very minor ethics concerns

**Final Justification:**

I thank the authors for their detailed and thoughtful rebuttal. I appreciate the effort they put into addressing the key concerns raised in the initial review, especially given the tight rebuttal timeline.

In particular, I find the newly added motion decoding and grayscale video reconstruction experiments in Appendices E and F to be valuable additions. These results help validate the claim that motion information is present in the recorded dorsal stream activity, even if full video-level reconstruction remains an open challenge. The authors also made appropriate revisions to clarify the limitations of comparing across different datasets, and I appreciate the more measured framing in the updated manuscript. I would also like to thank Reviewer PE1U for their thoughtful clarification regarding the frame length 200 ms.

While I still believe that future work with longer sequences and richer motion analyses would strengthen the utility of this dataset even further, the current contribution is already a significant step forward. Overall, I believe it will be a useful benchmark for the community moving forward.

For these reasons, I am raising my score to 5 (Accept).

**Limitations Weaknesses:**

1. Motion is under-explored. Only one frame of each 200 ms clip is reconstructed; this sidesteps the core claim that dorsal neurons encode motion. A video-level decoder (or at least optical-flow analyses) would better test the hypothesis.

2. Non-comparable reconstruction metrics – PSNR/LPIPS are reported for dorsal (video-frame) and ventral (static TVSD images) on different stimulus domains, undermining the “apple-to-apple” comparison.

3. 200 ms clips (5 frames) are short for many self-motion models; longer sequences would aid future studies.

**Strengths Contributions:**

1. STSBench increases dorsal-stream sample size by ~50× over prior MT datasets (45 units) and enables modern deep-learning benchmarks.

2. The baseline settings (3-D Gabor, ResNet-Kinetics, ResNet-Self-Motion and ImageNet 2-D controls) in the experiments are fair.

3. Depth ablations (1/3/5/7-layer CNNs) and learned-filter visualizations (Fig 4c) are insightful, illustrating where classical Gabor intuition holds or fails.

---

> ### Author Rebuttal · Authors · 2025-07-30
>
> We thank the reviewer for their thoughtful comments on our work. We have made the following revisions to address the reviewer’s critiques.
>
> **Comment 1:** *Motion is under-explored. Only one frame of each 200 ms clip is reconstructed; this sidesteps the core claim that dorsal neurons encode motion. A video-level decoder (or at least optical-flow analyses) would better test the hypothesis.*
>
> **Response 1:** We thank the reviewer for this suggestion. We have added new results from a motion direction decoder and a grayscale video reconstruction model to the appendix to address this. The new results demonstrate that the average motion direction of each video can be predicted from neural activity. We leave more comprehensive analyses of motion statistics to future studies.
>
> We have added the following sentence discussing this revision to the main text: *“To assess the information about motion contained in neural activity in STSBench, we report results from a motion direction decoder in Appendix E and grayscale video reconstruction model in Appendix F.”*
>
> The new appendices can be found at the end of the rebuttal. We report quantitative results and describe the figures in text, as images and videos cannot be included in the rebuttal.
>
> **Comment 2:** *Non-comparable reconstruction metrics – PSNR/LPIPS are reported for dorsal (video-frame) and ventral (static TVSD images) on different stimulus domains, undermining the “apple-to-apple” comparison.*
>
> **Response 2:** We agree with the reviewer that this was not an “apples-to-apples” comparison. The comparison across datasets could be problematic due to differences in the naturalistic stimuli used as the reviewer noted, differences in neuron count as reviewer “daig” noted, and different definitions of a ‘neuron’ in the two datasets (spike-sorting vs threshold crossing). Consequently, we have revised the manuscript to focus primarily on the results from STSBench and refrain from drawing conclusions from comparisons between the two datasets. The relevant changes to the text are enumerated below:
> 1. Lines 12-15 now read, *“We show that our dataset can be used for benchmarking encoding models of dorsal stream neuronal responses and reconstructing visual input from neural activity.”*
> 2. In lines 69-70, contribution (iv) has been removed.
> 3. Lines 257-262 have been removed from the manuscript, and replaced by a summary of the reconstruction results and discussion of the limitations of comparing STSBench and TVSD: *“Taken together, the dorsal stream reconstruction results illustrate that the features of visual stimuli known to be represented by the dorsal stream can be recovered from neuronal activity in STSBench. The ventral stream reconstruction results validate our modeling approach and provide a qualitative reference, but direct comparisons between the datasets cannot be made due to differences in neuron counts, preprocessing techniques (spike-sorted in STSBench vs threshold-crossing in TVSD), and stimuli (natural videos in STSBench vs natural images in TVSD).”*
> 4. Lines 271-273 now read *“Our reconstruction results highlight the utility of STSBench for studying the features of the visual world represented by populations of neurons in the dorsal stream.”*
>
> **Comment 3:** *200 ms clips (5 frames) are short for many self-motion models; longer sequences would aid future studies.*
>
> **Response 3:** We agree that this would be an interesting direction for future studies. We chose to use 200 ms clips here to maximize stimulus diversity given a limited number of trials, but a dataset with longer sequences would nicely complement STSBench.
>
> **Comment 4:** *Lines 221-222: the statement that results “suggest limitations in our current understanding of MT/MST mechanistically as a Gabor filter bank and conceptually as a region optimized for self-motion estimation” is intriguing but brief. Could you elaborate on that?*
>
> **Response 4:**
> We thank the reviewer for the opportunity to elaborate on this. The original statement was referencing two complementary lines of work that have shaped our understanding of MT/MST. ‘Mechanistic’ referred to models that characterize MT responses as a weighted combination of input from V1 neurons (e.g., Nishimoto and Gallant 2011). ‘Conceptual’ referred to more recent work (e.g., Mineault et al. 2021) that demonstrates that MT/MST responses can be predicted by neural networks optimized for self-motion estimation tasks, providing a normative explanation for why MT/MST neurons do what they do.
>
> Our original statement suggested that the superior performance of the 3D CNN-5 model trained end-to-end compared to the 3D Gabor and 3D ResNet - Self Motion models highlights limitations of existing models. However, as reviewer “daig” noted, this may have been too strong a claim, since there are other models in the literature that could also be tested. We have therefore revised the sentence to make a more conservative claim: *“This result suggests that while models based on hand-tuned Gabor filter banks [26] or filters optimized for self-motion estimation [28] provide solid baselines, they may not capture responses in MT/MST as well as models trained end-to-end.”*
>
>
> **New Appendices**
>
> **Appendix E: Decoding optic flow**
>
>
> To assess whether neural activity in STSBench encodes motion information, we trained a regression model to predict the direction of motion in each video from neural activity. Motion direction labels were computed by applying the Farneback algorithm (Farneback 2003) to estimate dense optical flow between adjacent video frames. The resulting flow vectors were averaged over both space and time, then normalized to obtain a unit vector representing the motion direction for each video (Figure 11a).
>
>
> The input to the decoder was the vector of firing rates of all neurons in STSBench associated with a particular video. The output of the decoder was a prediction of the x and y values of the motion direction vector (Figure 11a). The RidgeCV ridge regression model in scikit-learn was used as the decoder. The model was trained on the STSBench training set, then evaluated on the held-out test set by measuring the cosine similarity between the predicted and true motion direction vectors. As a control, we trained the same model with randomly shuffled targets and averaged results over five shuffles. The model achieved a mean cosine similarity of 0.923 on the test set compared to 0.051 for the shuffled control (Figure 11b), indicating that motion direction can be robustly decoded from neural activity.
>
>
> To examine the dependence of decoding performance on the number of neurons used as predictors, we trained the decoder with subsets of the total population. We observed a linear relationship between the logarithm of the number of neurons and cosine similarity that began to saturate at ~100 neurons (Figure 11b).
>
>
> To identify the neurons that most strongly influenced the decoder output, we computed a feature importance score for each neuron, defined as the sum of the absolute values of its weights for predicting the x and y components of the motion direction vector. We then examined the firing rates of the eight most informative neurons as a function of video motion direction. The neurons exhibited clear Gaussian-like tuning to motion direction (Figure 11c), underscoring the potential utility of STSBench for studying how the dorsal stream processes motion.
>
>
> *Figure 11. Decoding optic flow from neural activity in STSBench. (a) Diagram of optic flow calculation and decoding for an example test set video. Dense optical flow is computed using the Farneback method across adjacent frames, then averaged over time and space to assign a single motion direction vector to each video. A linear regression model is trained to predict the motion direction vector from neural activity in STSBench. (b) Decoding performance (cosine similarity) on the held-out test set as a function of the number of neurons used for decoding. The horizontal line denotes the performance of a target-shuffled control. (c) Feature importance analysis. The normalized firing rates of the top eight features (neurons) in the decoder are plotted as functions of motion direction.*
>
>
>
> **Appendix F: Grayscale video reconstruction**
>
> To further assess the presence of motion information in neural activity in STSBench, we trained a model to reconstruct five-frame grayscale videos conditioned on neural activity. The VQ-VAE and diffusion model architecture and training procedure were identical to the image reconstruction models, except that the first and last layers of the VQ-VAE were adapted to process inputs and outputs of shape [5, W, H] rather than [3, W, H], replacing the color channels with a time dimension. For evaluation, we report the cosine similarity between the average motion direction in the ground truth video and its reconstruction, computed as in Appendix E.
>
>
> The reconstruction model achieved a cosine similarity of 0.438 between the average motion direction of the reconstructed and ground truth videos, substantially higher than chance (0 if motion direction is uniformly distributed; 0.0485 with the circular mean of the training set). This is well below the cosine similarity of 0.923 obtained in Appendix E when decoding motion direction directly from neural activity, suggesting that there is considerable room for improving upon these results. Example reconstructions are illustrated in Figure 12 and corresponding videos are included in the code repository.
>
> *Figure 12. Reconstructing grayscale videos from neural activity in STSBench. Four example ground truth videos and corresponding reconstructions are displayed. Arrows denote the mean optic flow direction for adjacent frames computed using the Farneback method in either the ground truth video (red) or reconstructed video (blue).*

---

> > ### Comment · Reviewer_L7sH · 2025-08-02
> > **Thanks for the additional experiments and rebuttal**
> >
> > I thank the authors for their detailed and thoughtful rebuttal. I appreciate the effort they put into addressing the key concerns raised in the initial review, especially given the tight rebuttal timeline.
> >
> > In particular, I find the newly added motion decoding and grayscale video reconstruction experiments in Appendices E and F to be valuable additions. These results help validate the claim that motion information is present in the recorded dorsal stream activity, even if full video-level reconstruction remains an open challenge. The authors also made appropriate revisions to clarify the limitations of comparing across different datasets, and I appreciate the more measured framing in the updated manuscript. I would also like to thank Reviewer PE1U for their thoughtful clarification regarding the frame length 200 ms.
> >
> > While I still believe that future work with longer sequences and richer motion analyses would strengthen the utility of this dataset even further, the current contribution is already a significant step forward.  Overall, I believe it will be a useful benchmark for the community moving forward.
> >
> > For these reasons, I am raising my score to 5 (Accept).

---

### Official Review · Reviewer_PE1U · 2025-07-01

**Rating:** 5
**Confidence:** 4

**Summary:**

The study addresses a critical gap in NeuroAI by providing large-scale neural recordings from the dorsal visual stream (MT/MST), an area less studied compared to the ventral stream. Additionally, the authors benchmark multiple 3D CNN models, revealing current limitations in dorsal stream alignment. Finally, the authors develop a reconstruction pipeline using diffusion models conditioned on neural activity patterns, enabling extraction of visual features encoded in the neural population responses.

**Dataset Code Accessibility:**

Yes

**Dataset Code Comments:**

Dataset is well-structured and accessible through Kaggle, and the full code base is included in the supplementary material. Training details are specified in the Appendix.

**Ethical Considerations:**

No, there are no or only very minor ethics concerns

**Final Justification:**

All my points has been addressed and I am more confident in my review after the rebuttal from the authors. Additional thoughts can be found in another comment of mine and I will keep my score for acceptance.

**Limitations Weaknesses:**

I do not find major weaknesses or limitations. There are only a few minor comments/questions.

1. In Fig 1b & Fig 3a, the scales of "True Rate" and "Pred. Rate" are different. Please consider using the same scale so it is clear whether some of the predictions overshoot or undershoot.

2. In Fig 3a, $W_f$ seems to be a flattened vector, but in Section 4.1 Readouts, $w^{f}_{i,j}$ seems to be an element of a 2D matrix. If I understand correctly, the weights that map CNN activations to neuronal response can be factorized into two tensors of shape $c \times d \times 1$ and $1 \times w \times h$. Please consider making this more consistent.

I have read the paper multiple times, and I do not have other comments or questions.

**Strengths Contributions:**

Key strengths include:

1. Providing a major dataset (with nearly 50-fold increase over existing ones) with 2,244 STS neurons (with multiple cell types) recorded during natural video viewing, enabling more extensive modeling of dorsal stream responses. Since most brain-ANN comparisons have focused on the ventral stream due to data availability, extending this approach to the dorsal stream is natural, timely, and pivotal. This offers significant relevance to NeuroAI and huge potential impacts.

2. Stimuli are naturalistic instead of parametric.

3. Higher resolution of neuronal responses (single-neuron level) compared to fMRI recording for the dorsal stream.

The work bridges the ventral-dorsal data imbalance and enables deeper investigation of motion/spatiotemporal processing in primates by developing ANN models that better align with brain. The reconstruction approach also offers an effective way to decode visual features from neural activity, allowing us to better understand what features are encoded in dorsal stream areas.

Writing is very clear; figures are accurate and informative. Code is available in the supplementary materials.

---

> ### Author Rebuttal · Authors · 2025-07-30
>
> We thank the reviewer for their careful reading and thoughtful evaluation of our work. We appreciate the reviewer's positive assessment of our manuscript as timely and pivotal. Below, we address each of the reviewer’s comments and describe the corresponding revisions made to the manuscript.
>
>
> **Comment 1:** *"In Fig 1b & Fig 3a, the scales of "True Rate" and "Pred. Rate" are different. Please consider using the same scale so it is clear whether some of the predictions overshoot or undershoot."*
>
> **Response 1:** We appreciate the reviewer’s suggestion. In the revised version of Fig. 1b and Fig. 3a, we have adjusted the y-axis scale so that “True Rate” and “Pred. Rate” are now plotted on the same scale for each neuron.
>
> **Comment 2:** *In Fig 3a, $W_f$ seems to be a flattened vector, but in Section 4.1 Readouts, $w^f_{i,j}$ seems to be an element of a 2D matrix. If I understand correctly, the weights that map CNN activations to neuronal response can be factorized into two tensors of shape  $ c \times d \times 1 $ and $ 1 \times w \times h$. Please consider making this more consistent.*
>
> **Response 2:** We thank the reviewer for this helpful observation. The reviewer’s understanding of the readout mechanism is correct, and our illustration of $W_f$ as a flattened vector in Fig. 3a was confusing. We have now revised Fig. 3a to illustrate that $W_f$ is a 2D matrix, consistent with the notation in Section 4.1.
>
> **Additional Notes:** In response to a comment from reviewer "L7sH", we have added motion decoding and video reconstruction results to the appendix. The text of the new appendices can be found at the bottom of our response to reviewer "L7sH".

---

> > ### Comment · Reviewer_PE1U · 2025-08-02
> > **Thanks for the rebuttal and additional thoughts**
> >
> > I thank and applaud the authors' efforts for making sufficient and solid responses regarding all reviewers' comments within the limited time window. All my questions have been addressed and I'm more confident in keeping my score (i.e. 5, accept).
> >
> > Regarding the comments from Reviewer L7sH, I do understand there might be certain constraints in comparsions across different datasets collected using different methodology due to different research goals. But from my perspective, the decoding experiments primarily serve as a validation of dataset which basically suggests "we indeed can decode something useful which we already knew, so the data we collected is not completely noise". The decoding performances themselves may not be as essential as the dataset itself. But indeed I believe the authors have made reasonable responses regarding this point.
> >
> > Also, indeed 200 ms might be insufficient for many self-motion tasks, yet this dataset still constititues a great contribution for simple self-motions. In comparison to neural data for ventral stream studies (e.g. IT, average firing rate within 70-170 ms), I think 200 ms for dorsal stream is not unreasonable as a step forward to bridge the current data imbalance in dorsal/ventral stream datasets.

---

### Official Review · Reviewer_daig · 2025-07-03

**Rating:** 5
**Confidence:** 4

**Summary:**

The authors presented STSBENCH, a large-scale dataset of neuronal recordings from the superior temporal sulcus (STS), collected while monkeys viewed natural videos.

**Dataset Code Accessibility:**

Partly

**Dataset Code Comments:**

I was only able to access the neural data from the provided link; the video clips appear to be missing. Please clarify whether they are available elsewhere.

**Ethical Considerations:**

No, there are no or only very minor ethics concerns

**Limitations Weaknesses:**

In Figure 5, the dorsal stream is shown to capture low spatial frequency luminance contours, while the ventral stream captures detailed information about form and object identity. Could this difference be partly due to the number of neurons involved in the reconstruction across the two datasets?

Some statements appear overstated. For example, line 211: "This suggests limitations in our current understanding of MT/MST mechanistically as a Gabor filter bank." This claim seems overly strong; few researchers would apply the Gabor filter bank framework to higher visual areas such as MT or MST, beyond early visual cortex (V1/V2).

Overall, this is a valuable dataset that will likely be useful to the field.

**Strengths Contributions:**

There have been several established benchmarks for the ventral visual stream, such as BrainScore, MacaqueITBench, and the Things Ventral Stream Dataset (TVSD). This new benchmark of the dorsal stream will significantly advance our understanding of dorsal stream function and support the development of new encoding and reconstruction models for this pathway.

The dataset is clearly described and easy to understand.

---

> ### Author Rebuttal · Authors · 2025-07-30
>
> Response: We thank the reviewer for their thoughtful comments on our work. We have made the following revisions to address the reviewer’s critiques.
>
> **Comment 1:** *In Figure 5, the dorsal stream is shown to capture low spatial frequency luminance contours, while the ventral stream captures detailed information about form and object identity. Could this difference be partly due to the number of neurons involved in the reconstruction across the two datasets?*
>
> **Response 1:** We thank the reviewer for this question. The comparison across datasets could be problematic due to differences in the number of neurons as the reviewer noted, differences in the type of naturalistic stimuli used as reviewer “L7sH” noted, and different definitions of a ‘neuron’ in the two datasets (spike-sorting vs threshold crossing). Consequently, we have revised the manuscript to focus primarily on the results from STSBench and refrain from drawing conclusions from comparisons between the two datasets. The relevant changes to the text are enumerated below:
> 1. Lines 12-15 now read, *“We show that our dataset can be used for benchmarking encoding models of dorsal stream neuronal responses and reconstructing visual input from neural activity.”*
> 2. In lines 69-70, contribution (iv) has been removed.
> 3. Lines 257-262 have been removed from the manuscript, and replaced by a summary of the reconstruction results and discussion of the limitations of comparing STSBench and TVSD: *“Taken together, the dorsal stream reconstruction results illustrate that the features of visual stimuli known to be represented by the dorsal stream can be recovered from neuronal activity in STSBench. The ventral stream reconstruction results validate our modeling approach and provide a qualitative reference, but direct comparisons between the datasets cannot be made due to differences in neuron counts, preprocessing techniques (spike-sorted in STSBench vs threshold-crossing in TVSD), and stimuli (natural videos in STSBench vs natural images in TVSD).”*
> 4. Lines 271-273 now read: *“Our reconstruction results highlight the utility of STSBench for studying the features of the visual world represented by populations of neurons in the dorsal stream."*
>
> **Comment 2:** *Some statements appear overstated. For example, line 211: "This suggests limitations in our current understanding of MT/MST mechanistically as a Gabor filter bank." This claim seems overly strong; few researchers would apply the Gabor filter bank framework to higher visual areas such as MT or MST, beyond early visual cortex (V1/V2).*
>
> **Response 2:** We appreciate the reviewer’s feedback regarding our description of the encoding model results. We agree that Gabor filter bank models are not typically viewed as complete models of MT/MST, and have revised line 211 accordingly: *“This result suggests that while models based on hand-tuned Gabor filter banks [26] or filters optimized for self-motion estimation [28] provide solid baselines, they may not capture responses in MT/MST as well as models trained end-to-end.”*
>
> We have also revised our conclusions to emphasize the potential utility of our dataset for refining models of MT/MST and to avoid implying that our current results are sufficient to do so. Lines 269-271 now read: *“Our encoding results demonstrate that simple 3D CNNs trained end-to-end outperform other baseline models of MT/MST, underscoring the potential of STSBench to refine models of dorsal stream visual processing.”*
>
> **Comment 3:** *I was only able to access the neural data from the provided link; the video clips appear to be missing. Please clarify whether they are available elsewhere.*
>
> **Response 3:** We thank the reviewer for this clarification question. We have confirmed that the video clips are contained in the Kaggle dataset, linked on lines 101-102 of the manuscript, in a directory named dorsal_stream/. The clips can be viewed by navigating to the link and downloading the dataset.
>
> **Additional Notes:** In response to a comment from reviewer "L7sH", we have added motion decoding and video reconstruction results to the appendix. The text of the new appendices can be found at the bottom of our response to reviewer "L7sH".

---

> > ### Comment · Reviewer_daig · 2025-08-05
> > **comments**
> >
> > Thank you for addressing my concerns; they have been fully resolved. This is a good and potentially useful dataset. However, I have a follow-up question, as I am still not entirely certain in which respect it can be useful. Although the authors presented good decoding results based on static visual information, might it be more useful for detecting motion-related or “where”-related visual information, given the nature of the classical dorsal pathway?

---

> > > ### Author Response · Authors · 2025-08-05
> > > **Response to comments**
> > >
> > > We thank the reviewer for this insightful question. Indeed, we anticipate that our dataset will be useful for exploring how motion and space are represented in the dorsal stream. Towards this end, we added results from a motion direction decoder and video reconstruction model in our response to reviewer “L7sH”.  Our reconstruction results provide a starting point for assessing the presence of visuospatial information in the dorsal stream, and we encourage future studies to explore this direction further.

---

> > > > ### Comment · Reviewer_daig · 2025-08-05
> > > > **reply**
> > > >
> > > > That sounds great.

---

### Decision · Program_Chairs · 2025-09-18

**Decision:**

Accept (poster)

**Comment:**

This paper introduces STSBench, a large-scale single-neuron dataset recorded with Neuropixels while animals viewed thousands of short natural video clips. All reviewers agree on accepting the paper. The reviewers acknowledge that this is a ~50× scale-up over prior dorsal datasets, which addresses a long-standing bottleneck and will shape future dorsal-stream modeling work. The paper presented fair baselines and depth ablations. After carefully reading the paper, rebuttal, and reviews, the AC agrees with  the reviewers on accepting the paper.

===== FINAL UPDATE FROM DB Track PCs ====

The final decision for this paper has been taken by the program chairs after consultation with the SACs. All Senior Area Chairs have ranked papers according to the feedback from the AC during the review process. We decided to leave the original meta-review to reflect the opinion of the AC in light of the initial discussions with reviewers and SAC.